Multi-walled carbon nanotubes produced after forest fires improve germination and development of Eysenhardtia polystachya

Juárez-Cisneros Gladys 1
Gómez-Romero Mariela 2
Reyes de la Cruz Homero 1
Campos-García Jesús 1
Villegas Javier vmoreno@umich.mx 1
1 Instituto de Investigaciones Químico Biológicas, Universidad Michoacana de San Nicolás de Hidalgo , Morelia , Michoacán , México
2 Cátedras CONACYT-Facultad de Biología, Universidad Michoacana de San Nicolás de Hidalgo , Morelia , Michoacán , México
de Azevedo Jr. Walter
Electronic publication date: 2020 Apr 23
Publication date: 2020
Volume: 8
Electronic Location ID: e8634
Received 2019 Oct 8; Accepted 2020 Jan 26
Copyright: ©2020 Juárez-Cisneros et al.
Copyright year: 2020
Copyright holder: Juárez-Cisneros et al.
License: This is an open access article distributed under the terms of the Creative Commons Attribution License, which permits unrestricted use, distribution, reproduction and adaptation in any medium and for any purpose provided that it is properly attributed. For attribution, the original author(s), title, publication source (PeerJ) and either DOI or URL of the article must be cited.
License URL: https://creativecommons.org/licenses/by/4.0/

Keywords: Natural multi-walled carbon nanotubes, Nanomaterials, Plant growth promotion, Amorphous carbon

Funding: C.I.C./UMSNH CONACYT This research was funded by C.I.C./UMSNH grant and a doctoral fellowship grant by CONACYT to Gladys Juárez-Cisneros. The funders had no role in study design, data collection and analysis, decision to publish, or preparation of the manuscript.

==============================
Background

Multi-walled carbon nanotubes (MWCNTs) are nanoparticles with countless applications. MWCNTs are typically of synthetic origin. However, recently, the formation of MWCNTs in nature after forest fires has been documented. Previous reports have demonstrated the positive effects of synthetic MWCNTs on the germination and development of species of agronomic interest; nevertheless, there is practically no information on how synthetic or natural MWCNTs affect forest plant development. In this report, based on insights from dose-response assays, we elucidate the comparative effects of synthetic MWCNTs, amorphous carbon, and natural MWCNTs obtained after a forest fire on Eysenhardtia polystachya plant.

Methods

E. polystachya seeds were sown in peat moss-agrolite substrate and conserved in a shade house. Germination was recorded daily up to 17 days after sowing, and plant development (manifested in shoot and root length, stem diameter, foliar area, and root architecture parameters) was recorded 60 days after sowing.

Results

The treatments with natural MWCNTs accelerated the emergence and improved the germination of this plant, thus while untreated seeds achieve 100% of germination within 16th day, seeds supplemented with natural MWCNTs at doses of 20 µg/mL achieve the above percentage within the 4th day. Natural MWCNTs also promoted fresh and dry biomass in all applied treatments, specially at doses of 40 µg/mL where natural MWCNTs significantly promoted leaf number, root growth, and the dry and fresh weights of shoots and roots of seedlings. Seeds supplemented with doses between 20 and 40 µg/mL of amorphous carbon achieving 100% of germination within the 6th day; however, seeds supplemented either with doses of 60 µg/mL of the above carbon or with synthetic MWCNTs at all the tested concentrations could achieve at most 80 % and 70% of germination respectively within the 17 days. Finally, neither treatments added with amorphous carbon nor those added with synthetic MWCNTs, showed significant increases in the fresh and dry biomass of the tested plant. Likewise, the survival of seedlings was reduced between 10 and 20 % with 40 and 60 µg/mL of amorphous carbon, and with synthetic MWCNTs in all the doses applied was reduced at 30% of survival plants.

Conclusions

These findings indicate that MWCNTs produced by wildfire act as plant growth promoters, contributing to the germination and development of adapted to fire-prone conditions species such as E. polystachya.

Introduction

Multi-walled carbon nanotubes (MWCNTs) are nanoparticles with unique physicochemical properties that have recently been the focus of scientific, commercial, and biotechnological interest (De Volder et al., 2013; Zhu et al., 2013). In the last two decades, the applications of MWCNTs in different plant species of agronomic interest have been explored. The results documented so far show that MWCNTs promote plant growth. The capacity of MWCNTs to promote early emergence of seeds and increase the percentage of germination has been demonstrated in corn (Tiwari et al., 2014), soybean, barley, and corn hybrids (Lahiani et al., 2013). It has also been reported that synthetic MWCNTs promote elongation and root branching in Brassica oleracea, Daucus carota, Cucumis sativus, Allium sp. (Cañas et al., 2008), and Cicer arietinum (Tripathi, Sonkar & Sarkar, 2011). However, the phytotoxic effects of MWCNTs have also been reported in several plant species (Vithanage et al., 2017). For example, in lettuce (Lactuca sativa L.) (Ikhtiari et al., 2013), MWCNTs inhibited germination, and limited growth and biomass by inducing cell death. Similarly, in tomato and spinach, single-walled carbon nanotubes (SWCNTs) were shown to inhibit radical elongation (Cañas et al., 2008), while in Cucurbita pepo L., exposure to MWCNTs significantly decreased the germination percentage, root and shoot length, and biomass accumulation (Hatami, 2017). Contrasting effects of these nanoparticles have been associated with intrinsic characteristics, such as their shape, dimensions, electrical conductivity, stability, and limited solubility (Scown, Van Aerle & Tyler, 2010), as well as the concentration of nanoparticles and the plant species used as the test model (Jackson et al., 2013). To date, MWCNTs have been considered to be synthetic nanoparticles (Liu et al., 2014), obtained principally by arc-discharge, laser ablation, and chemical vapor deposition methods (Zaytseva & Neumann, 2016). However, Lara-Romero et al. (2017) demonstrated the presence of MWCNTs with ∼10 layers of graphene in the calcined wood of resinous pine species after forest fire events. Furthermore, the authors performed a thermogravimetric analysis (TGA) to determinate the amount of MWCNTs in the burned wood. The analysis revealed that the wood samples of Pinus oocarpa contained ∼2.8% (w/w) of these nanomaterials. These findings raise questions about the eco-physiological impacts of natural MWCNTs on the plant populations of these ecosystems. There is practically no information about the effects of MWCNTs on indigenous plant populations; nevertheless, these nanoparticles may play a significant role in the growth and development of such plant species.

Eysenhardtia polystachya is a leguminous shrub, characteristic of pine forests in Mexico subjected to fire disturbance. Owing to it is rapid growth and abundant seed production, it is an interesting candidate to test the effects of MWCNTs. The objective of this study was to evaluate and compare the effects of amorphous carbon and MWCNTs of natural and synthetic origin on the germination and morphological seedling variables of E. polystachya.

Materials & Methods

MWCNTs and amorphous carbon specifications

Synthetic MWCNTs used in this study had an outer diameter of 6–13 nm, the internal diameter of 2.0–4.0 nm, length of 2.5–20 µm, an average wall thickness of 7–13 graphene layers, and purity >98% (Aldrich).

Natural MWCNTs were obtained from carbonized P. oocarpa wood samples collected six weeks after a forest fire in Huashan mountain in Nahuatzen Michoacán, Mexico, as described by Lara-Romero et al. (2017). The samples were first sieved using a 0.2-micron mesh to homogenize the particle size, and then calcined at 620 °C for three hours to mineralize up to 98% of organic matter from amorphous sources (amorphous carbon).

Non-crystalline carbon samples from Pinus montezumae (rich in amorphous carbon) were also collected from the same site and at the same time, as mentioned previously.

Nanomaterial solutions were prepared by adding natural MWCNTs, synthetic MWCNTs, and amorphous carbon individually to sterile distilled water. For each nanomaterial, solutions with three different concentrations: 20, 40, and 60 µg/mL were prepared. These solutions were sonicated to facilitate the carbon material dispersion, 60 min before the seed treatments. The above concentrations were chosen in the range of 10–60 µg/mL, based on previous studies Lara-Romero et al. (2017), that used synthetic MWCNTs (with structural features similar to those found in the natural samples) to evaluate growth and development E. polystachya.

Seed germination and plant growth

Seeds of E. polystachya were collected from Cerro del Punhuato, Michoacán, Mexico. Seeds were disinfected with 10% (v/v) H2O2 for 20 min in Brandson 5510 sonicator. Subsequently, each seed was planted in a polypropylene container with peat moss (PREMIER®)-agrolite substrate (1:2) that had been previously sterilized (Gómez-Romero et al., 2013). A total of 1.0 mL of the suspension containing the carbon materials at the prepared concentrations were then added to the seeds. The experiments were performed using a completely randomized experimental design. Treatments consisted in: (I) natural MWCNTs, (II) synthetic MWCNTs, and (III) amorphous carbon, each with three different levels (concentration: 20, 40, 60 µg/mL), replicated eight times in polypropylene containers each with one seed. Treatments were compared with the control (concentration: 0 µg/mL of carbon source supplemented) with the same number of seeded polypropylene container replicates.

The seeded containers were then placed in a shade house, and watered three times a week, maintaining field capacity during the experiment.

Treatments were evaluated at 18 different time intervals; germination was recorded daily up to 17 days after sowing, and plant development was recorded at the end of the trial, i.e., 60 days after sowing.

To record its development, plants were removed from the containers, and the roots were washed with running water to remove the adhering substrate residues. The percentage of survival was registered, after which the plants were cut from the base of the stem, and shoot and root length, stem diameter, and foliar area were measured. Variables of root architecture, such as primary root length, lateral roots, tertiary roots, and root volume, were also recorded using the WinRHIZO software coupled to an EPSON Expression 11000XL scanner (Régent Instruments Inc., Québec, CA). Finally, the shoot and the root were weighed separately, then placed in paper bags and allowed to dry at room temperature before being weighed again to obtain the dry weight.

Statistical analysis

Germination cumulated data, available for 17 days, were analyzed using a generalized linear model (GLM) with a binomial distribution and Cox analysis, to determine the behavior of the germination curves between treatments over time.

Growth data were analyzed using one-way ANOVA, and the means were compared using Tukey’s tests with P < 0.05, in GraphPad software. The analyses were performed using eight repetitions to balance out the effect of non-germinated seeds.

Results

Seed germination and survival of E. polystachya

Natural MWCNTs accelerated the germination of this legume; at the end of the germination test, Cox’s proportional hazards test indicated that the germination rates during the test period were significantly different (X2 = 17.04, P = 0.01). E. polystachya seeds exposed to different carbon sources showed different germination rates. Three days after sowing, 60–90% germination was recorded in seeds treated with natural MWCNTs compared with 40% those kept as control. While six days after sowing, seeds treated with natural MWCNTs had reached 100% germination in all the doses applied, compared with 90% of germination in control, an 80%–100% germination in seeds treated with amorphous carbon and 70–80 with synthetic MWCNTs (Table 1). Furthermore, the control seeds took 16 days to reach 100% germination, and it was evident that synthetic MWCNTs slowed down seed germination, which reached a maximum of 90% in the same period.

Table 1 Effect of synthetic MWCNTs, carbon amorphous and natural MWCNTs on Germination and survival of Eysenhardtia polystachya.

Seeds of E. polystachya were supplemented with 1.0 mL suspension containing either 0 (control), 20, 40, or 60 µg/mL of the different carbon materials. Germination was recorded daily up to 17 days after sowing, and survival was recorded at the end of the trial, 60 days after sowing. The results represent the mean of three independent assays with n = 8. The germination was analyzed through a generalized linear model (GLM) for the data, with a binomial distribution and a Cox analysis.

Treatment	Carbon source (µg/seed)	Days after planting	Survival (%)	
		3	4	5	6	14	15	16	17		
		% of germination		
Control	0	40	70	80	90	90	90	100	100	100	
Natural MWCNTs	20	60	80	90	100	100	100	100	100	100	
40	90	100	100	100	100	100	100	100	100	
60	90	90	100	100	100	100	100	100	100	
Amorphous carbon	20	40	60	80	100	100	100	100	100	100	
40	50	70	80	100	100	100	100	100	90	
60	50	70	70	80	80	80	80	80	80	
Synthetic MWCNTs	20	20	70	70	70	70	70	70	70	70	
40	50	50	80	80	80	90	90	90	70	
60	60	60	80	80	80	80	80	80	70	

E. polystachya plant observed sixty days after sowing (Table 1), showed 100% survival in the control group and groups treated with natural MWCNTs (all doses) or 20 µg/mL of amorphous carbon. In contrast, seeds treated with 40 and 60 µg/mL of amorphous carbon showed 90% and 80% survival, respectively, indicating that an increase in amorphous carbon concentration resulted in a decreased survival percentage. The addition of synthetic MWCNTs also negatively affected E. polystachya survival. We obtained 70% survival with all the doses applied of synthetic MWCNTs.

Aerial growth of E. polystachya

The effects of natural MWCNTs, amorphous carbon, and synthetic MWCNTs at concentrations of 0, 20, 40, and 60 µg/mL on the seeds of E. polystachya grown in shade house conditions sixty days after sowing are shown in the Figs. 1 and 2. We observed that treatment with 40 µg/mL of natural MWCNTs significantly promoted leaf formation, when compared with treatment with synthetic MWCNTs and control (Fig. 2A), but no significant difference was observed in other treatments (Tukey test with P < 0.05). Furthermore, treatments containing natural MWCNTs significantly increased the foliar area at all concentrations tested, while amorphous carbon and synthetic MWCNTs did not have any significant effect (Fig. 2B). In addition, no significant differences were observed in the height of E. polystachya plants treated with natural MWCNTs or amorphous carbon and those kept as controls (Fig. 2C) according with Tukey test (P < 0.05). However, treatments with synthetic MWCNTs negatively affected plant height at concentrations of 60 µg/mL. The aerial dry weight of plants treated with 40 µg/mL of natural MWCNTs was significantly higher, while plants under other treatments did not show any difference with respect to the control (Fig. 2D).

Figure 1 Images showing the effect of synthetic MWCNTs, carbon amorphous and natural MWCNTs on growth of Eysenhardtia polystachya.

Seeds of E. polystachya were planted in containers with peat moss-agrolite substrate and supplemented with 1.0 mL suspension containing either 0 (control), 20, 40, or 60 µg/mL of the different carbon materials. A–L and M–X correspond to plants after 20 and 60 days of growth, respectively.

Figure 2 Effect of synthetic MWCNTs, amorphous carbon and natural MWCNTs on aerial biometric parameters of Eysenhardtia polystachya plants.

Seeds of E. polystachya were supplemented with 1.0 mL suspension containing either 0 (control), 20, 40, or 60 µg/mL of the different carbon materials. After 60 days of planting the plants were harvested and biometric variables were recorded. (A) Leaves number, (B) foliar area, (C) height, (D) aerial dry weight. Bars represent mean ± SE of three independent assays. n = 8. One-way analysis of variance (ANOVA) was carried out with Tukey’s post hoc test; statistical significance (P < 0.05) between treatments with respect to control is indicated with different lowercase letters.

Root architecture of E. polystachya

The effects of natural and synthetic MWCNTs and amorphous carbon on root architecture of E. polystachya were evaluated 60 days after sowing (Fig. 3). It was observed that the primary root length showed significant increases in treatments with natural MWCNTs, compared to the control plants (Fig. 4A); however, the number of secondary roots did not show significant differences between the treatments containing the tested materials and the control (Fig. 4C). It was evident that treatments with 40 and 60 µg/mL of natural MWCNTs modified the root architecture by promoting the formation of tertiary roots (Fig. 4B), significantly increases in the root volume were observed in plants treated with 40 and 60 µg/mL of natural MWCNTs compared to the control group and treatments containing synthetic MWCNTs or amorphous carbon (Fig. 4D) according to with Tukey test (P < 0.05).

Figure 3 Effect of natural MWCNTs, amorphous carbon and synthetic MWCNTs on root development of Eysenhardtia polystachya.

The images show root architecture changes in response to different carbon materials in E. polystachya roots harvested 60 days after planting. (A–D) Natural MWCNTs. (E–H) Amorphous carbon. (I–L) Synthetic MWCNTs.

Figure 4 Effect of synthetic MWCNTs, amorphous carbon and natural MWCNTs on root architecture of Eysenhardtia polystachya plants.

Seeds of E. polystachya were supplemented with 1.0 mL suspension containing either 0 (control), 20, 40, or 60 µg/mL of the different carbon materials. After 60 days of planting the plants were harvested and root architecture variables were recorded. (A) Primary root length, (B) lateral roots number, (C) tertiary roots number, (D) root volume, (E) root fresh weight, and (F) root dry weight. Bars represent mean ± SE of three independent assays. n = 8. One-way analysis of variance (ANOVA) was carried out with Tukey’s post hoc test; statistical significance (P < 0.05) between treatments with respect to control is indicated with different lowercase letters.

Furthermore, the fresh and dry root weights of E. polystachya seeds treated with natural MWCNTs at concentrations higher than 40 µg/mL were significantly increased (Figs. 4E and 4F) compared to the weights recorded in other treatments. Conversely, the addition of amorphous carbon and synthetic MWCNTs significantly decreased the dry root weight at concentrations above 20 and 40 µg/mL according to with Tukey test (P < 0.05).

Discussion

The use of synthetic MWCNTs as plant growth promoters has been reported in several crop plants in the two last decades (Khodakovskaya et al., 2012; Khodakovskaya et al., 2013; Lahiani et al., 2015). The scientific findings report both positive (Joshi et al., 2018a; Joshi et al., 2018b) and negative (Ikhtiari et al., 2013; McGehee et al., 2017) effects of synthetic MWCNTs on plants species. However, to date, the effects of naturally occurring MWCNTs are poorly known. Thus, in the present study, we evaluated the effects of natural and synthetic MWCNTs as well as amorphous carbon on the germination and development of E. polystachya plants grown in shade house conditions.

The responses of this legume to the MWCNTs treatments were contrasting, depending on the origin of the nanomaterial, i.e., MWCNTs of natural origin collected from forest fires events promoted early emergence and increased the germination percentage of the seeds, while synthetic MWCNTs negatively affected seed germination (Table 1). It has been previously reported that the effects of MWCNTs in plants and other organisms depend on their physicochemical properties, such as surface area, length, and diameter, the presence of functional groups, load, shape, and solubility.

In this study, the MWCNTs formed naturally after forest fires lead to better tested plant growth and development than MWCNTs obtained from chemical synthesis. It has been shown that MWCNTs with different characteristics affect seed germination. Early germination induced by synthetic MWCNTs has been reported in tomato seeds, soybean, barley, corn (Lahiani et al., 2013), oat (Joshi et al., 2018b), wheat (Wang et al., 2012; Joshi et al., 2018a), and Lupinus elegans (Lara-Romero et al., 2017). Increased seed germination has been associated with increased water uptake during seed imbibition, facilitated by the formation of new pores during penetration of seed coat and cell walls by the MWCNTs; however, the action mechanism of these structures on seed germination is not completely clear. In that context, it has been documented that several chemical and physical factors can influence the biochemical and physiological events that control the germination in seeds (Nelson et al., 2012; Asghar et al., 2017).

The effect of MWCNTs has also been documented in other physiological stages of plant development. It has been suggested that a plant response to these nanomaterials depends on their intrinsic chemical characteristics, concentration (Lahiani et al., 2013; Lara-Romero et al., 2017), dispersion method (Joshi et al., 2018a; Joshi et al., 2018b), and also on the plant species (Zhai et al., 2015; Zaytseva, 2016) and the experimental conditions in which it develops (Tiwari et al., 2014). Thus, the effects of MWCNTs can be positive, as observed in the E. polystachya plants cultivated with 40 µg/mL of natural MWCNTs, where the plants showed greater vegetative area, more abundant foliage, and more aerial area. Our results evidenced that natural MWCNTs modified the root architecture of this legume, as a higher number of tertiary roots and higher root volume were observed, which is beneficial for its establishment, allowing for greater gaseous exchange and absorption of water and minerals (Lynch, 1995). In addition, plants treated with natural MWCNTs showed a significant increase in dry weights of both shoot and root. Similar effects have been documented for synthetic MWCNTs in oat (Joshi et al., 2018b), wheat (Joshi et al., 2018a), corn (Tiwari et al., 2014; Zhai et al., 2015), and Lupinus elegans (Lara-Romero et al., 2017). However, the mechanisms by which MWCNTs promote plant growth and development are not clear. Some reports suggest that MWCNTs activate mechanisms of cell division (Khodakovskaya et al., 2012) and promote elongation of xylem and phloem cells, which consequently influence the uptake of water and nutrients (Joshi et al., 2018a; Joshi et al., 2018b).

It must be noted that toxic effects of synthetic MWCNTs on species of agronomic interest have also been previously reported, such as in Lactuca sativa (Ikhtiari et al., 2013), Amaranthus tricolor L., and Cucumis sativus (Begum, Ikhtiari & Fugetsu, 2014). In this study, we found that synthetic MWCNTs, at the concentrations tested, negatively affected the physiological development of E. polystachya, by altering germination, morphometric variables aerial plant parts, and root architecture. The mechanisms associated with MWCNT toxicity have not been elucidated in detail; however, they are associated with cell death in roots and leaves, caused by an increase in the generation of reactive oxygen species (Ikhtiari et al., 2013) and rupture of cell membranes (Begum, Ikhtiari & Fugetsu, 2014).

Conclusions

In this work, for the first time, we report the effects of natural MWCNTs collected from burned trees after a forest fire. We observed that these MWCNTs improved and accelerated germination in E. polystachya seeds and promoted growth, in both aerial and underground parts. We also observed that amorphous carbon did not significantly affect the development of this plant. In contrast, MWCNTs from synthetic origins were observed to negatively affect plant development. These results suggest that natural nanoparticles produced after forest fires may positively affect the growth and development of plants in these ecosystems.

Supplemental Information

Supplemental Information 1 Document of sampling area in Nahuatzen Michoacán

Click here for additional data file.

Supplemental Information 2 Raw dat of germination parameters and Biometric parameters

Germination Excel tab shows Eysenhardtia seed responses, in terms of germination parameters per seed, to different concentrations of MWCNTs. Biometric Excel tab shows root and aboveground responses per plant, in terms of structure and biomass, to different concentrations of MWCNTs

Click here for additional data file.

Additional Information and Declarations

Competing Interests

Author Contributions

Data Availability

Jesús Campos-García is an Academic Editor for PeerJ.

Gladys Juárez-Cisneros performed the experiments, prepared figures and/or tables, authored or reviewed drafts of the paper, obtained the suspensions with the different carbon materials, and approved the final draft.

Mariela Gómez-Romero performed the experiments, analyzed the data, prepared figures and/or tables, authored or reviewed drafts of the paper, developed the statistical analysis, and approved the final draft.

Homero Reyes de la Cruz analyzed the data, authored or reviewed drafts of the paper, and approved the final draft.

Jesús Campos-García conceived and designed the experiments, analyzed the data, authored or reviewed drafts of the paper, and approved the final draft.

Javier Villegas conceived and designed the experiments, authored or reviewed drafts of the paper, and approved the final draft.

The following information was supplied regarding data availability:

The raw measurements are available in the Supplemental Files.

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
