# Peer review of "Multi-walled carbon nanotubes produced after forest fires improve germination and development of Eysenhardtia polystachya"

_PeerJ, doi:10.7717/peerj.8634_

## Round 0.1 · original submission · Minor Revisions

Three specialists in the field evaluated the present manuscript, and they have minor concerns related to this paper. The reviewers have described all points that should be answered by the authors. Considering the evaluation carried out by the reviewers, I recommend minor revision in this submission.

·

Basic reporting

For me the paper is clear but I not English mother tongue.

Introduction and literature are satisfying.

Structure is conform to journal standards.

Figures are of good quality and interesting. Caption and figures label need minor corrections (see attached PDF).

Raw data was supplied for germination and plant growth

Experimental design

The research fits the Aims and Scope of the journal.

The research fills a knowlege gap on the role of natural fire products on post-fire germination.

Investigation was conducted rigorously.

Methods are reproducible. For clarity I suggest minor revision of experimental design.

Validity of the findings

In M&M I suggest to clarify experimental design (page 7 lines 103-104) as follows:

treatments are three: natural MWCNTs, synthetic MWCNTs and amorphous carbon, each with three different levels (concentration: 20, 40, 60 mg/mL) replicated ten times in polypropylene container each containing 1 seed (?). Treatments where compared with the control (concentration: 0 mg/mL) with the same number of seeded polypropylene container replicates.

Additional comments

The work analysis the role of two natural wildfire products, MWCNTs and amorphous carbon, and one synthetic product, MWCNTs, on germination and seedling biometric growth parameters of a native legume species (E. p.) growing in fire prone pine forests. Natural wildfire products were collected from two Mexican pine species in the same burnt forest.
Results indicate that natural MWCNTs improved significantly emergence and growth of seedling of the tested species, mainly at 40 g/mL of concentration.
Overall, the manuscript is of interest and contains remarkable data for a large audience, the methods applied are rigorous, well explained and the results obtained are confirmative.
In the attached PDF are my specific and minor comments.

Reviewer 2 ·

Basic reporting

The manuscript " Multi-walled carbon nanotubes formed after forest fires improve germination and development of Eysenhardtia polystachya " is interesting and well written and experiments were well performed. Literature used is appropriate. The presentation of the data is clear.

However, In the Introduction some information about environmental concentrations of MWCNTs should be introduced. If there are no information about it, at least indicate that no information on environmental concentrations exists and why…. I know that is difficult to quantify, but it is important to mention it.

Experimental design

Experiments were well performed. However, it should be added some important details regarding the experimental design:

Line 94 – The choice of the concentrations are based on which premise? Previous works? or concentrations found in the environment?

Lines 103-104: “The experiments were performed using a completely randomized experimental design using ten treatments with n = 8.” Here, when you say n=8 you mean 8 replicates per treatment? In table 1 I saw that you have 3 assays and 8 seeds per assay.
in this statement is not clearer this information. please clarify and indicate that you have 3 replicates per treatment.

Validity of the findings

Results are convincing and conclusions are well stated, linked to the aim.

Additional comments

General Comment: The manuscript " Multi-walled carbon nanotubes formed after forest fires improve germination and development of Eysenhardtia polystachya " is interesting and well written and experiments were well performed. The presentation of the data is clear. Results are convincing and publication is recommended. I still have some comments in order to improve the manuscript before to be published.

Lines 158 – 159 - treatments with synthetic MWCNTs negatively affected plant height for all concentrations?

Table 1 – can the authors include the standard deviation? Can you also include significative differences among treatments (at least for day 17). For survival percentage, please introduce the number of days after planting (60?)

Figure 1 A and B – It is possible to maintain the same order for both 1A and 1B (for example: control, synthetic MWCNTs, Natural MWCNTs and Amorphous carbon)? I suggest to maintain the same order used in figure 2. Plants from control seems to be smaller (aereal part) when compared with remaining conditions. However, this data is not concordant with data presented in fig 2C, were is possible to see that in mean, the height of control plants, for example, was significantly higher than plants exposed to synthetic MWCNTs.

figure 2 – statistical significance is indicated with different uppercase letters in the figure and not with lowercase letters as referred by the authors in the figure caption. It is missing the letter “C” in the figure 2C.

Figure 3 – it would be interesting to have also a figure of roots from control.

Figure 4 – figure 4D does not have the letters of statistical significance; again uppercase letters were used for the remaining figures, instead of lowercase as mentioned in figure caption.

Reviewer 3 ·

Basic reporting

Comments attached

Experimental design

Comments attached

Validity of the findings

Comments attached

Additional comments

Comments
Multi-walled carbon nanotubes formed after forest fires improve germination and development of Eysenhardtia polystachya (#41130)
Author reported the formation of MWCNTs in nature after forest fires has been documented and their effect on Eysenhardtia polystachya seeds germination. Results showed that natural MWCNTs in all applied doses accelerated the emergence and improved the germination of this plant, significantly promoting leaf number, root growth, and the dry and fresh weights of shoots and roots was studied. These findings indicate that MWCNTs from natural sources act as plant growth promoters, contributing to the germination and development of forest species such as E. polystachya.
This work is interesting, which is a significant advancement over existing knowledge, but it needs minor improvements before considering for publication.
Specific comments
1. Abstract should contain some quantitative information also.
# What about the toxicity of nanoparticle to the tissues, have authors considered this factor ?
# the abbreviation used must be explained on their first appearance, or provide separate list of abbreviations
# How author will compare their findings with physical pre-sowing seed treatment because these are safe versus chemical treatment ?, some references are included below to discuss this issue in present investigation
Journal of Photochemistry and Photobiology B: Biology, 173 (2017) 344-352., Journal of Photochemistry and Photobiology B: Biology, 166 (2017) 212-219., Journal of Photochemistry and Photobiology B: Biology, 165 (2016) 283-290., Biocatalysis and Agricultural Biotechnology, 6 (2016) 176-183., Journal of Photochemistry and Photobiology B: Biology, 170 (2017) 314-323., Effects of X-ray irradiation on growth physiology of Arachis Hypogaea (Var. Kampala). , Chemistry International 3(2017) 296-300.

---

## Round 0.2 · accepted · Accept

The authors carried out all modifications indicated by the reviewers. In my view, the manuscript can be accepted as it is.